# A Controlled, Retrospective, Single-Center Study to Evaluate the Role of a Probiotic Mixture Administered during Pregnancy in Reducing Streptococcus Agalactiae Swab Positivity and the Frequency of Premature Rupture of Amniochorionic Membranes

**DOI:** 10.3390/microorganisms12101979

**Published:** 2024-09-30

**Authors:** Maurizio Arduini, Elena Laurenti, Massimiliano Cazzaniga, Alexander Bertuccioli, Ilaria Cavecchia, Mariarosaria Matera, Nicola Zerbinati, Francesco Di Pierro

**Affiliations:** 1Obstetrics and Gynecology Department, S. Maria della Misericordia Hospital, 06129 Perugia, Italy; maurizio.arduini@ospedale.perugia.it (M.A.); elena.laurenti@tiscali.it (E.L.); 2Scientific & Research Department, Velleja Research, 20125 Milano, Italy; maxcazzaniga66@gmail.com; 3Microbiota International Clinical Society, 10123 Torino, Italy; alexander.bertuccioli@uniurb.it (A.B.); ilaria.cavecchia@osp-koelliker.it (I.C.); jajamatera74@gmail.com (M.M.); 4Department of Biomolecular Sciences, University of Urbino Carlo Bo, 61122 Urbino, Italy; 5Microbiomic Department, Koelliker Hospital, 10134 Turin, Italy; 6Department of Paediatric Emergencies, Misericordia Hospital, 58100 Grosseto, Italy; 7Department of Medicine and Technological Innovation, University of Insubria, 21100 Varese, Italy; nicola.zerbinati@uninsubria.it

**Keywords:** precision probiotics, *E. faecium*, pregnancy, intrapartum antibiotic

## Abstract

Intrapartum antibiotic prophylaxis, considered able to prevent streptococcal transmission from mother to newborn and its severe negative consequences, leads to microbiota dysbiosis, described as having a negative impact on well-being in both elements of the dyad. *Enterococcus faecium* L3 is a probiotic strain capable of exerting strong antagonistic activity against most streptococci, including *S. agalactiae*, due to the production of bacteriocins (known as enterocins A and B). A proprietary probiotic mixture containing the strain L3 demonstrated, in 2016, a significant reduction in episodes of PROM in pregnant women, with a less-than-expected effect on the vaginal–rectal presence of the pathogen *S. agalactiae*. With the aim of confirming the role exerted by the probiotic mixture in PROM episodes and to better understand the value of its impact on the clinical detection of *S. agalactiae*, we have retrospectively analyzed the results obtained in 125 L3-treated (over 12 weeks) women versus 125 untreated controls. Despite some limitations, our analysis has confirmed the role exerted by the probiotic in significantly reducing the following: (1) episodes of PROM, (2) vaginal–rectal positivity for *S. agalactiae,* and (3) the need to administer intrapartum antibiotics for prophylaxis. It likely also suggests operating using a cultural method very specific to *S. agalactiae* when testing women who were administered an *Enterococcus*-based probiotic.

## 1. Introduction

*Streptococcus agalactiae*, also known as group B *Streptococcus* (GBS), is a Gram-positive bacterium that is commonly found in the gut and vaginal mucosa of humans. Typically an asymptomatic colonizer, GBS can spread to other peripartum tissues, such as the placental–fetal unit during pregnancy, the neonate during labor and delivery, or the mammary glands during lactation. In these environments, GBS can cause serious disease, including chorioamnionitis, preterm labor, fetal death, neonatal sepsis or meningitis, and clinical mastitis [1,2,3]. In developed countries, GBS is the leading cause of severe neonatal infection and about one-third of newborns of carrier women are colonized at the time of delivery. During the first 7 days of life, about 3% of colonized infants may develop an early-onset infection that can be fatal or can induce serious consequences [4].

In 1996, the Centers for Disease Control (CDC) in Atlanta published the first guidelines for the prevention of streptococcal disease. These were updated in 2002 and 2010 [5]. The method adopted in Italy, based on the Guidelines on Physiological Pregnancy drawn up in 2011, is based on vagino-rectal screening performed between weeks 35 and 37 of gestation, with intra-partum antibiotic treatment only for women who have tested positive [6]. Several randomized clinical trials have shown that intrapartum intravenous antibiotic prophylaxis in women with streptococcus, performed with penicillin G or penicillin A, starting at the beginning of labor and continuing until the time of delivery, leads to a significant reduction in the risk of early infection [7].

As reported by the Association of Italian Hospital Obstetricians and Gynecologists (AOCOI) in its recent report [8], despite recent advances in the clinical and therapeutic fields, in the last thirty years, the incidence of preterm birth (PTB) in Western countries has not decreased, resulting in the commitment of a significant amount of economic and social resources. This is due to diagnostic difficulties and to the lack of causal treatment resulting from limited knowledge of the risk factors, etiology and pathogenetic mechanisms responsible. PTB is closely linked to the premature or preterm premature rupture of amniochorionic membranes (PROM and pPROM, respectively) [9].

The overall incidence of preterm birth reported is approximately 12–13% in the United States and 5–9% in Europe, with a slightly lower rate of birth at 32 weeks, at approximately 2% [10]. In Italy, the most recent data, updated to 2005, report for PTB a global value of 6.5%, which drops to 0.85% for births before 32 weeks [11]. According to modern conception, the so-called “PTB syndrome” is considered the result of a chronic process of multifactorial origin with heterogeneous manifestations in which the following factors are involved and interact differently with each other: (i) genetic predisposition (polymorphisms, abnormal allogeneic recognition of the fetus, allergic-type reactions); (ii) uterine environmental factors (infection, ischemia, hyperdistention, cervical insufficiency); (iii) inflammatory mechanisms; and (iv) endocrine disorders [12].

The causal role played by infection and/or inflammation for spontaneous PTB is well-documented. Indeed, 25–40% of PTB cases have an infectious cause, with a higher rate the earlier the gestational age at the time of clinical manifestation (preterm labor, PROM and pPROM). The microorganisms most frequently detected in amniotic fluid belong to Mycoplasmataceae, mainly *U. urealyticum*, followed by *S. agalactiae*, *E. coli*, *Fusobacterium* spp. and *G. vaginalis* [13,14,15,16]. They colonize the intrauterine compartment (deciduous, amniocorial membranes, amniotic fluid, placenta, cord and fetal tissues) predominantly via an ascending route from the vagina, but also via the transplacental hematogenous route (pneumonia, pyelonephritis, asymptomatic bacteriuria, appendicitis, periodontal disease), retrograde from the peritoneum through the fallopian tubes and are iatrogenic because of an invasive prenatal diagnosis procedure [17]. Infection alone may not be sufficient to cause PTB, which is often the result of a complex interaction between the microbial environment and the host’s immune and inflammatory response, including cytokines and prostaglandins [18].

Asymptomatic bacteriuria has a prevalence of 2–10% in pregnancy and is also associated with an increased risk of PTB [19]; low-to-moderate-quality evidence suggest that treatment with antibiotics results in a reduction in the incidence of low birth weight and preterm birth, which justifies screening practices with only a single urine culture in the first trimester [20]. Moreover, evidence support the correlation between bacterial vaginosis (BV), a term used to define a change in the vaginal ecosystem, and spontaneous PTB, with more than twice the increased risk [21]. Even if antibiotic therapy is described to counteract BV [22], its use, associated with a reduced incidence of PTB and low weight in preterm infants, remains very controversial and requires further deep investigation, as demonstrated by the different findings obtained using clindamycin or metronidazole [23].

However, antibiotic use in pregnancy is not without risk, as they can, in turn, cause alterations in the vaginal microbiota and increase the risk of miscarriage [24]. It is worth noting that any antibiotic treatment, whether topical or systemic, can lead to an alteration of the ratio between Gram-positive and Gram-negative bacteria in favor of the Gram-negative [25,26,27]. Due to LPS-linked inflammatory processes, their increase leads to systemic endotoxemia with enhanced production of inflammatory cytokines and prostaglandins, which, if early-activated, could promote the onset of PROM and PTB, as was experimentally observed [28]. Therefore, it seems logical, at least in theory, to intervene in a different way by trying to prevent intestinal and/or vaginal dysbiosis with tools such as bacterial therapy and “precision” probiotics [29], which have demonstrated, for example, peculiar anti-pathogenic properties [30].

The strain *E. faecium* L3 has been shown to exert a specific experimental anti-*S. agalactiae* role due to the peculiar release of two bacteriocins known as Enterocin A and Enterocin B [31,32,33]. Its clinical use in women during their 3rd trimester of pregnancy indeed demonstrated, in addition to a high safety profile and an anti-gut dysbiosis prophylactic effect, (i) a 6% decrease in the incidence of streptococcal colonization; (ii) an approximate 30% decline in episodes of PROM; (iii) fewer caesarean sections during labor; and (iv) a reduction in pathological umbilical cord blood pH [34]. Furthermore, the difference between the reduction of *S. agalactiae* observed by the swab-derived culture and the reduction in PROM episodes was too wide and maybe unexpected. The authors hypothesized that the use of the L3 strain, described as highly colonizing [34], could have influenced the results of the detection of *S. agalactiae* since the test used had shown, under certain cultural circumstances, to be unable to distinguish with high specificity the presence of *S. agalactiae* from the presence of *Enterococcus* spp. [35,36]. Since the results of this clinical investigation have never been replicated by any author since 2016, the aim of our non-profit study was to confirm the results obtained in that study, taking care to better balance the study groups (treated vs. controls) and deepen the analyses performed for *S. agalactiae* positivity.

## 2. Materials and Methods

### 2.1. The Study

This report concerns a retrospective, observational, controlled, single-center, non-profit, open-label study performed to evaluate the safety and efficacy of a specific probiotic mixture administered daily and orally in pregnant women from 24 to 36 weeks of gestation. The study was approved by the Ethics Committee for Human Experimentation (CESU) at the University of Urbino “Carlo Bo”, as reported in an extract of the minutes of session no. 79 on 22 February 2024. The trial has been registered at www.clinicaltrials.gov and the identifier is NCT06231056.

### 2.2. The Inclusion and Exclusion Criteria

The study retrospectively analyzed 250 pregnant women with a history of recurrent genitourinary infections, defined as at least three episodes within the previous year (details in Appendix A). Recurrent urinary tract infections (rUTI) were diagnosed using urine cultures [37], while bacterial vaginosis was identified through Amsel’s criteria [38] and/or Gram-staining with microscopy techniques [39]. Trichomonas infections were detected by nucleic acid amplification testing [39] and aerobic vaginitis was diagnosed based on clinical signs and vaginal pH evaluation [40]. Additionally, functional bowel discomfort—such as bloating, flatulence, constipation, or diarrhea—was assessed according to the Rome IV criteria [41], with symptoms reported for at least six months before enrollment. All 250 women were seen at our department between June 2019 and June 2020. The exclusion criteria were as follows: age under 18; presence of neurological, cardiac, pulmonary, hepatic, or renal disease; diagnosis of a serious metabolic disorder; history of cancer; use of probiotics and/or prebiotics during pregnancy (other than the prescribed probiotic); and refusal to sign the informed consent and privacy forms.

### 2.3. Study Endpoints

The primary endpoints of the study were the following: (i) bacteriological assessment (control of bacteriuria by urine culture and GBS positivity investigation by vaginal–rectal swab); (ii) clinical assessment at the end of pregnancy (PROM); (iii) need for induction of labor; and (iv) type of delivery (natural or caesarean section). The secondary endpoints of the study were the following: global assessment of antibiotic use (including asymptomatic and symptomatic bladder infection, GBS positivity, evidence of PROM and cesarean section), evaluation of tolerability, compliance and incidence of side effects.

### 2.4. The Protocol

Of the 312 patients affected by recurrent genitourinary infections who were considered enrolled and screened from a group of 424 women (Appendix A) attending our clinics between June 2019 and June 2020, 250 were, according to our inclusion and exclusion criteria, considered for analysis. Among them, 125 were treated with the probiotic mixture orally with a single dose per day (1 sachet after breakfast or after lunch); while 125 were not treated and were considered controls. The method of retrospective assignment to the probiotic group was either the consequence of a specific request made by the patient during the first visit (i.e., the patient expressly requested to be treated with a probiotic for preventive purposes), or based on the patient’s acceptance of the proposal of the physician who had examined her, which was made in relation to intestinal symptoms reported by the patient to be present in at least the last 6 months preceding the visit. The reported symptoms had to be attributable to functional and non-organic disorders. In doubtful cases, this aspect was defined by a gastroenterological visit. Conversely, if the patient did not request a probiotic approach or did not complain, during their first visit, of particularly relevant intestinal symptoms observed in the months immediately preceding the visit, the patient was not suggested to use the probiotic. Each patient was visited a minimum of 5 times during pregnancy and at least 2 times after delivery. Starting from the first visit, throughout the pregnancy and for the following trimester, each patient had the option to contact the department to have an immediate comparison with the physicians responsible.

### 2.5. The Tested Probiotic Product

The probiotic mixture retrospectively evaluated in our observational trial (iNatal^®^) is a nutritional supplement notified to the Italian Ministry of Health by Pharmextracta (Pontenure, Italy), according to the provisions of law No. 169 of 2004, on 4 September 2014 (notification number: 71640) and contains the following strains: (i) *Enterococcus faecium* L3 LMG P-27496 at a concentration of 10 × 10^9^ colony-forming units/dose (CFU/dose); (ii) *Bifidobacterium animalis* ssp. *lactis* BB-12 DSM 15954 at a concentration of 3 × 10^9^ CFU/dose; (iii) *Lactobacillus casei* RO215 CNCM I-3429 at a concentration of 3 × 10^9^ CFU/dose; and *Lactococcus lactis* ssp. *lactis* SP 38 DSM 26868 at a concentration of 3 × 10^9^ CFU/dose. The reported CFU/doses are intended to be disposed on the expiry date of the product.

### 2.6. Urine Culture, Swab Test, Antibiotic Treatments and Induction of Labor

A urine analysis was performed in all women between the 36th and 37th week of pregnancy. An antibiotic was selected based on the bacterial sensitivity and history of allergy of the patients and was administered immediately after receiving the result. If possible, cefalexin was used (500 mg every 12 h for 7 days). A rectal-vaginal swab (Todd Hewitt CNA Regular locked Swab Kit; Copan Diagnostics, Murrieta, CA, USA) was taken between the 36th and 37th week of pregnancy to check for the presence of *S. agalactiae,* while also applying the method previously described [36] to avoid a false diagnosis of group B streptococci misidentified as *Enterococcus*. Since some women could have an allergy to penicillin and/or ampicillin, all patients whose swab was positive for *S. agalactiae* have been subjected to an antibiogram. GBS-positive women without an allergy have been treated intra-partum with ampicillin (50 mg/kg i. m.). In cases of PROM, an antibiotic (mainly ampicillin or erythromycin) was administered 18 h from rupture. Where possible, to avoid the use of antibiotics for PROM, labor induction is managed as soon as possible. Labor induction was triggered by dinoprostone in the form of a vaginal device with a controlled release of 10 mg at 0.3 mg/h (for further details, see Appendix A). In cases of caesarean section, to protect against the surgical insult, if the patient’s allergy profile was compatible, an intrapartum cephalosporin was administered as the first choice. In relation to the results of the urine and GBS tests, PROM events and need for caesarean section, the choice of type and dosage of antibiotic was made in accordance with notes from the Italian Medicines Agency (AIFA) relating to antibiotic therapy [42].

### 2.7. Sample Size Calculation and Statistical Analysis

Reports and clinical investigations highlight a prevalence of GBS positivity in Italy between 27% and 32% [34,35], and according to a recent meta-analysis, the prevalence of GBS positivity in Italy is reported to be between 25% and 31% [43]. To detect a decrease of 15% with 80% power and a type I error of 0.05, our analysis requires 121 participants in each trial arm (Appendix A). Equivalence between the two groups (treated and control) was determined using Fisher’s exact test and the two-tailed Wilcoxon–Mann–Whitney test. The difference between the two groups in terms of side effects, clinical outcomes and antibiotic use was determined using the two-tailed Wilcoxon–Mann–Whitney test. JMP 10 statistical software for macOS was used and statistical significance was set at 95%.

## 3. Results

Our study is the result of a retrospective analysis of data obtained within a group of 250 pregnant women, originally selected from a group of 312 women (the reasons for the exclusion of 62 potential patients are reported in Appendix A) affected by recurrent genitourinary disorders and possible intestinal disorders. Of these 250 pregnant women, 125 were treated from the 24th to the 36th week of gestation with one dose per day of a specific probiotic product while the other 125 women acted as untreated controls. Patients in both groups were then subjected to bacteriological tests (urine culture and vaginal–rectal swab) and to clinical observation aimed at evaluating the frequency of certain pregnancy outcomes (PROM, need for induction of labor, natural birth or birth by cesarean section).

### 3.1. The Bacteriological and Clinical Outcomes

The specific features of the 250 women considered eligible for our retrospective analysis are reported in Table 1 and demonstrate that the two groups (A: probiotic; B: control) are comparable for all the parameters considered and not significantly different. As shown in Figure 1A and in Appendix A, 34 patients in the probiotic group and 37 in the control group demonstrated bacteriuria (*p* = 0.39). In addition (Figure 1B), a significant difference (*p* = 0.0163) was observed in vaginal–rectal swab positivity for *S. agalactiae*, with 34 patients in the probiotic group and 51 in the control group demonstrating positivity. Moreover, the PROM results were significantly different (*p* < 0.001) between the two groups, affecting 10 women in the probiotic group and 30 women in the control group (Figure 1C). Nearly significant (*p* = 0.0771), induction of labor was necessary in 14 women in the probiotic group and in 23 in the control group (Figure 1D). Lastly, a non-significant difference (*p* = 0.255) was observed with regard to cesarean sections. These were performed for 20 women in the probiotic group and 25 from the control group (Appendix A), while 105 women in the probiotic group and 100 in the control group delivered naturally.

### 3.2. The Women Treated with Antibiotics

In relation to the various outcomes measured and the clinical conditions of the different patients, it was then possible to evaluate the number of cases in the two groups in which it was necessary to administer an antibiotic. As shown in Table 2, 89 patients of the group treated with the probiotic and 115 patients of the control group were treated with the antibiotic, with a significant difference (−29.2%).

### 3.3. The Tolerability, Compliance and Incidence of Side Effects

The secondary endpoints of the study were tolerability, compliance and incidence of side effects. Probiotic tolerability was reported to be “very good” or “good” by 118 patients and “acceptable” by 7 patients. Compliance was demonstrated to be quite high, since all 125 women in the probiotic group declared to have been adherent to no less than 95% of the probiotic doses. Side effects (Appendix A) observed in the probiotic group were overlapping and not significantly different, both in type and severity, from those observed in the control group, demonstrating, therefore, no specificity.

## 4. Discussion

Our study retrospectively evaluated both the microbiological aspects (urine culture and positivity to *S. agalactiae*) and the possible pathological outcomes found in pregnancy (PROM) after having used, in 50% of the sample analyzed (N = 125), a specific probiotic that has previously demonstrated protective effects [34]. The total sample of pregnant women enrolled (N = 250) had in common a previous history of recurrent genitourinary infections.

The treatment, which lasted 12 weeks, determined both a significant reduction in positivity for *S. agalactiae* (27.2% vs. 40.8%) and in PROM episodes (8% vs. 24%). Conversely, we observed a non-significant reduction both in urine culture positivity (27.2% vs. 29.6%) and in need for cesarean section (16% vs. 20%) and a near-significant reduction with regard to the need for labor induction (11.2% vs. 18.4%). With regard to *Streptococcus*, we went from about 40% in the control group to about 27% in the treated group, with a reduction of 13%. The value observed in the control group is high compared to the Italian average (reported to be up to 30%) but the sample enrolled was selected to have the historical characteristics of recurrent infection. We believe that our selection may have influenced the high values observed in the control group. With regard to global antibiotic use, in the group treated with the probiotic formula the number of women administered an antibiotic was about 30% less. These results not only confirmed the general lines of the results previously obtained in 2016 [34], but also allowed us to highlight a stronger correlation compared to what has been previously observed between the presence of *S. agalactiae* and PROM episodes. Having adopted a microbiological method [36] capable of avoiding the possible cultural misinterpretation described between *Enterococcus* spp. and *S. agalactiae* has, in fact, allowed us to improve the precision of the results. It is likely that the misinterpretation is numerically irrelevant in standard conditions. Differently, in conditions of prolonged administration of a strain of *Enterococcus* described to be highly colonizing, the protocol itself significantly risks altering the obtained results. We did not perform a differential analysis aimed at intercepting how many false positives we would have obtained with the standard culture method; in any case, in our study, the results obtained relating to *S. agalactiae* were significant (a 13.6% reduction versus the control in our retrospective analysis, versus a 6% reduction versus the control in the 2016 study). A significant direct consequence linked to a lower positivity for *S. agalactiae* is a significantly lower recourse for intrapartum antibiotic therapy (for details on the protocol, see Appendix A), with a more eubiotic impact on the microbiota of the mother and newborn [44,45,46]. PROM at the end of pregnancy but before the onset of labor involves as a more serious consequence, an increase in the risk of both maternal and fetal infection. In cases of PROM, it is therefore necessary to evaluate the possibility for antibiotic therapy and induction of labor based on the presence of clinical and anamnestic risk factors and the outcome of the bacteriological screening for group B beta-hemolytic *Streptococcus*. Of course, the chance offered by the induction of labor has reduced by 90%, in both groups, the need for antibiotic therapy (the antibiotic treatment starts 18 h after the rupture of the membrane). Moreover, the observed reduction in PROM has allowed us to reduce the use of antibiotics. Overall, considering the results of the urine tests, GBS analyses, number of PROM episodes and the need to intervene with a cesarean section procedure, the use of antibiotics in the probiotic group involved 26 fewer women than in the control group (Table 2).

The tested probiotic proved to be a tool characterized by an excellent safety profile, determining good tolerability and compliance. Furthermore, the side effects, those unexpected with respect to the patient’s history, were found to be similar in type and severity to those observed in the untreated subjects. However, since this was not one of the objectives of our retrospective analysis, we did not collect data on the disappearance of any particular intestinal disorder using specific questionnaires. We are, therefore, unable to say whether the probiotic has played a eubiotic role at the gut level, capable, for example, of counteracting and/or normalizing historical situations of functional constipation or diarrhea. At least, in our study’s conditions, the probiotic has not significantly changed the frequency of pathogens detected at the urinary level, therefore, we cannot highlight any eubiotic properties exerted in the bladder microbiota.

Our study has numerous limitations, since it is not a prospective study, nor is it performed in a blind and/or placebo-controlled manner. Furthermore, our analysis evaluates the impact of a probiotic on a selected sample (with recurrent genitourinary infections) of patients who are not necessarily identical to the “standard” sample of pregnant women. Finally, most of the treated patients, in addition to having a history of recurrent genitourinary infections, like the control group, also showed functional gastrointestinal problems. The two groups cannot, therefore, be considered identical in terms of clinical history. Despite these limitations, this study has nevertheless allowed us to confirm, after many years, the validity of the results obtained previously in a completely different center and with different clinicians and authors, obtaining, at the same time, a likely more precise result with regard to GBS vaginal–rectal detection and antibiotic use.

Based on the data obtained, we are currently organizing a double-blind, placebo-controlled trial that can finally quantify the impact the administration of this probiotic has on reducing positivity for *S. agalactiae* (and, therefore, on the need to operate with intrapartum antibiotics), and on the number of PROM and pPROM episodes (in our study, we did not observe any cases of pPROM, even in the controls; therefore we are not able to affirm anything on this matter on the basis of the results).

## 5. Conclusions

Despite some limitations, our retrospective analysis has shown that the use of a specific probiotic mixture containing, among others, the strain *E. faecium* L3, described to be anti-*Streptococcus, aside from being* well-tolerated, significantly reduces (1) the vaginal–rectal swab positivity to *S. agalactiae*, (2) the episodes of PROM, and (3) the need to operate on the mother–newborn couple with intrapartum antibiotic prophylaxis.

## Figures and Tables

**Figure 1 microorganisms-12-01979-f001:**
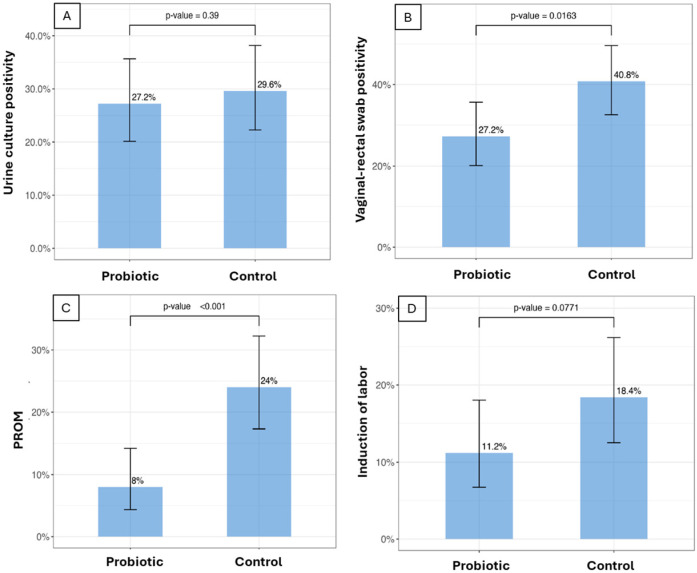
The bacteriological and clinical outcomes in women of the two groups. Urine cultures (**A**), vaginal–rectal swabs (**B**), episodes of PROM (**C**) and the need for labor induction (**D**) are shown. Significant differences are observed in *S. agalactiae* positivity (**B**) and episodes of PROM (**C**), demonstrating a likely protective effect exerted by the probiotic treatment.

**Table 1 microorganisms-12-01979-t001:** The features of the 250 analyzed patients. The numbers represent the patients.

	Group A (N = 125) Probiotic-Treated	Group B (N = 125) Control Group
Maternal age §	34.7 ± 2.8	33.9 ± 5.2
Maternal BMI (ante-pregnancy) §	24.3 ± 2.9	25.4 ± 3.1
Previous pregnancy	85	78
Previous abortion	7	9
Previous CS ^	25	19
Mother parity	Primiparous = 61	Primiparous = 68
Ethnicity (EU, AF, AS, Other) *	88, 19, 18, 0	83, 22, 13, 3
Education (P, M, H, U) °	15, 25, 44, 41	18, 22, 40, 45
Employment (Yes, No)	94, 31	90, 35
Marital status (Married, unmarried)	105, 20	109, 16

§ Values are expressed as an average ± standard deviation. ^ CS: cesarean section. * EU: Europe; AF: Africa; AS: Asia. ° P: primary school, M: middle school, H: high school; U: university. The differences observed in the values between the two groups are not significant.

**Table 2 microorganisms-12-01979-t002:** The number of cases in which it was necessary to administer an antibiotic in the two different study groups.

Reason for Antibiotic	Probiotic Group (P) Antibiotic Administered	Control Group (C) Antibiotic Administered	Δ (P vs. C)	*p*
Urine	34 out of 34	37 out of 37	−8.8%	N. S.
GBS	34 out of 34	51 out of 51	−50%	<0.01
PROM	1 out of 10	3 out of 30	−66%	N.S.
CS	20 out of 20	24 out of 24	−20%	N.S.
Total	89 out of 125	115 out of 125	−29.2%	<0.05

GBS: Group B *Streptococcus* (*S. agalactiae*); PROM: premature rupture of the membranes; CS: cesarean section; P: probiotic group; C: control group; vs: versus; N.S.: not significant.

## Data Availability

The raw data supporting the conclusions of this article will be made available by the authors on request.

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
