# Peer review of "A Controlled, Retrospective, Single-Center Study to Evaluate the Role of a Probiotic Mixture Administered during Pregnancy in Reducing Streptococcus Agalactiae Swab Positivity and the Frequency of Premature Rupture of Amniochorionic Membranes"

_microorganisms, 2024, doi:10.3390/microorganisms12101979_

Round 1

Reviewer 1 Report (Previous Reviewer 4)

Comments and Suggestions for Authors

Please find the comments attached below in a Word document.

Author Response

Reviewer 1

Dear and Esteemed Reviewer,

We have received your new review requests.

They seem, apparently, identical to those you sent us previously and to which we had already responded.

If we are in error, we apologize.

Below are your comments/requests and our responses.

Congratulations to the authors for addressing a very interesting issue, vital for the pregnancy outcome and neonatal health as well. Selection of patients, criteria and methods used are appropriate and explained in detail.

R: Tx a lot.

However, there are several things I would like to clarify:

Why patients with asymptomatic bacteriuria were not included since it's known it can cause PROM and preterm birth?

R: Patients with asymptomatic bacteriuria were not excluded at all exactly for the same reasons you reported (as mentioned by us in the Introduction section). All the urine coltures positive were treated (Fig. 1A). A precise statement (lines 148-150) has been added in Materials and Methods section.

Similarly, why only patients with bowel discomfort were included, thus limiting the benefits for only symptomatic patients?

Main inclusion criteria were: 1) pregnant and 2) be with a history of genitourinary infections. Some of them were also affected by functional gut disorders (that is not organic, like IBD). Particularly to them, according to our routine hospital practice, we proposed the use of the probiotic. For clarity, please see Supplementary File 1 (It is a scheme added in this new version). As you can see, we have at the end given the probiotics above all, but not only, to women who declared (besides to be pregnant and with a history of vaginal and urinary infection) to complin from function gut diseases. We kindly remind you that our analysis is retrospective. Therefore at the time of taking a decision concerning who take the product and who do not take it, the fact of having ALSO a gut discomfort prompted us to suggest the use of the probiotic. Anyway, ss you can see from Supplementary File 1, also women not having gut discomfort have been treated with the product.

In Figure 1, graphs for clinical outcomes for natural delivery and CS should be added.

R: The figure has been clearly and independently reported as Supplementary File 5. The way of delivery (natural versus CS) was not significantly different between the 2 groups and adding that graph to Figure 1 would reduce the size of the global figure and that of any single graph (they are already 4). We have seen that in terms of visibility the figure become less clear for the reader. Therefore our proposal, also because the result is “negative”, and this is clearly said inside the text, We kindly propose you to let that graph in the supplementary file. Of course, If for you is mandatory to add that graph in the Figure 1, we will try to arrange the Figure 1 in a different way. Thanks a lot for understanding.

Table 1. What was the criteria for antibiotic administration, e.g. why only 1 (P) and 3 (C) patients with PROM received antibiotic?

R: As reported in the manuscript, according to the Italian guidelines, the antibiotic in case of PROM has to be administered only if labor doesn’t start (naturally or by induction) within 18 hours. The difference in the table is therefore only because 1 woman in the P group and 3 women in the C group did not have a labor within 18 hours.

Also, why induction of labor and natural delivery groups aren't included in Table 1?

R: Now the Table you are referring to has become Table 2. Anyway, that table sum ups and reports only the cases where it was mandatory to administer a specific antibiotic. Its title is indeed: Table 2. Number of cases in which it was necessary to administer antibiotics in the two different study groups. Of course neither natural delivery nor labor induction foresee the use of antibiotics.

What was the discrimination criteria for labor induction instead of antibiotic therapy?

R: As said above, since you could wait 18 hours to decide if to administer an antibiotic in case of PROM, women close to the 18th hours, that do not need of CS and that were negative for GBS are normally subjected to labor induction to avoid the need of using the antibiotic.

I look forward to the revised version.

Tx. We have appreciated your review and comments.

The Authors

Reviewer 2 Report (Previous Reviewer 3)

Comments and Suggestions for Authors

Clarity is improved.  There appears to be a typo in the abstract, "cultural." 

Author Response

Reviewer 2

Clarity is improved.

R: Thanks a lot for your comment.

There appears to be a typo in the abstract, "cultural."

R: You are right. It has been changed.

Thanks a lot for all.

The Authors

This manuscript is a resubmission of an earlier submission. The following is a list of the peer review reports and author responses from that submission.

Round 1

Reviewer 1 Report

Comments and Suggestions for Authors

Dear Authors this work cannot be published due to a series of biases

1) lack of statistical data: sample size, clear definition of objectives. It ranges from genitourinary infections to prom, preterm birth, to complications of GBS infections

2) final results  suggest a success rates absolutely different   when compared to all literature data. this is related to the mis-design of your study

3) what do you mean when you define genitourinary disorder : urine? vaginitis? vaginosis ? what else? and which diagnostic methods? swab, nugget, etc ?No clear definition of recruited patients

necessary to review the entire study design

Comments on the Quality of English Language

Poor level of English in term of structure of phrase , terms and grammar

Reviewer 2 Report

Comments and Suggestions for Authors

I recommend rejecting this manuscript in accordance with the academic editor's recommendation.

Comments on the Quality of English Language

I recommend rejecting this manuscript in accordance with the academic editor's recommendation.

Reviewer 3 Report

Comments and Suggestions for Authors

*Interesting study as follow-up to earlier work of some of the authors on an important topic of probiotics to reduce Antenatal GBS, intrapartum antibiotic prophylaxis, PROM and GI symptoms. 

*Existing literature on probiotics to reduce antenatal GBS was not incorporated into literature review. Rather, the authors referred only to their prior works. 

*2.2 "study enrolled" is personification.

*2.4 The protocol reads as if you conducted a prospective study. "Retrospective assignment to probiotic group" is not clear. I think you were trying to indicated that the probiotic group participants received probiotics for several reasons. This whole paragraph needs more clarity for the reader as this does not read like a retrospective study.  Your two cohorts are really quite different based on probiotic group assignment. Some of the controls did not have a problem or request for treatment. 

*I would recommend that the demographics table be moved to the body of the main paper as is the style of most clinical trials.

*How can a retrospective observational study be controlled?

*The first paragraph of the findings beginning at line 3-6  is unclear and seems to contradict section 2.1.  Was this a primary study or a secondary analysis of another study?  Was that primary study published-if so be clear and provide reference?

*Were the urine cultures all taken at the same gestation? 

*How mode of birth is relevant outcome of probiotic use? Did you think they would differ significantly or are you reporting it as a general perinatal outcome?

*Findings about PROM are contradictory between abstract and discussion.  In the discussion you indicated that you could not draw conclusions about a reduction in the incidences of PROM since there were zero in both groups. In the abstract you indicated a significant reduction.  

*line 282 "more significant" is not recommended terminology.  Something is either significant or not. 

*lines 295-297  unclear.

*line 300 over statement/unclear statement "completely superimposable." Please rewrite for clarity. 

The discussion contains the generally needed pieces but contains many extraneous words that reduce clarity.  "Certainly" (line 306, line 309,"obvious limitations" (line 320), "results got (line 320)"

*Limitations need to include the variation in reasons received probiotics. Also clarity about the type of study this was (primary vs secondary analysis). 

*Line 315, unclear-"Conditions may exist for a randomized controlled trial"

*Conclusion line 322, "has shown once again" is confusing. 

Comments on the Quality of English Language

Extraneous words detract from clarity and give the impression the author/s seek drama over clarity and precision. 

Reviewer 4 Report

Comments and Suggestions for Authors

Congratulations to the authors for addressing a very interesting issue, vital for the pregnancy outcome and neonatal health as well. Selection of patients, criteria and methods used are appropriate and explained in detail. 

However, there are several things I would like to clarify:

Why patients with asymptomatic bacteriuria were not included since it's known it can cause PROM and preterm birth? Similarly, why only patients with bowel discomfort were included, thus limiting the benefits for only symptomatic patients?

In Figure 1, graphs for clinical outcomes for natural delivery and CS should be added.

Table 1. What was the criteria for antibiotic administration, e.g. why only 1 (P) and 3 (C) patients with PROM received antibiotic? Also, why induction of labor and natural delivery groups aren't included in Table 1?

In lines 291-293 what was the discrimination criteria for labor induction instead of antibiotic therapy?

I look forward to the revised version.